# Dissemination of Tylosin Residues in the Poultry Environment: Evaluating Litter and Droppings as Sources of Risk

**DOI:** 10.3390/antibiotics14050477

**Published:** 2025-05-08

**Authors:** María Belén Vargas, Ignacia Soto, Francisco Mena, Paula Cortés, Ekaterina Pokrant, Lina Trincado, Matías Maturana, Andrés Flores, Aldo Maddaleno, Lisette Lapierre, Javiera Cornejo

**Affiliations:** 1Laboratory of Food Safety, Department of Preventive Animal Medicine, Faculty of Veterinary and Animal Sciences, University of Chile, Santiago 8820808, Chile; maria.vargas.s@ug.uchile.cl (M.B.V.); francisco.mena@ug.uchile.cl (F.M.); katiavalerievna@uchile.cl (E.P.); 2Doctorate Program of Forestry, Agriculture, and Veterinary Sciences (DCSAV), University of Chile, Santiago 8820808, Chile; paula.cortes@ug.uchile.cl (P.C.); lina.trincado@ug.uchile.cl (L.T.); 3Laboratory of Veterinary Pharmacology (FARMAVET), Faculty of Veterinary and Animal Sciences, University of Chile, Santiago 8820808, Chile; ignacia.soto@ug.uchile.cl (I.S.); matias.maturana@ug.uchile.cl (M.M.); andres.flores@veterinaria.uchile.cl (A.F.); amaddaleno@veterinaria.uchile.cl (A.M.); 4Laboratory of Bacterial Pathogens Diagnostic and Antimicrobial Resistance, Department of Preventive Animal Medicine, Faculty of Veterinary and Animal Sciences, University of Chile, Santiago 8820808, Chile; llapierre@uchile.cl

**Keywords:** tylosin, residues, dissemination, poultry, litter, dropping

## Abstract

**Introduction:** Tylosin, a veterinary antimicrobial belonging to the macrolide family, is commonly used in the poultry industry. Residues generated from its use can be present in the litter and droppings of treated birds. Due to the diverse uses of poultry byproducts, such as fertilizing agricultural soils or incorporation into the diets of other animal species, there is a risk to public health, as the presence of antimicrobial residues favors the development of antimicrobial resistance, which is a global problem. **Objective:** This study aimed to evaluate the dissemination of tylosin residues from the litter and droppings of treated birds and untreated birds in a controlled broiler environment. **Methods:** Bird droppings and litter samples were collected and analyzed using HPLC-MS/MS to detect and quantify tylosin residues. **Results:** The residue concentrations detected in the dropping matrix only exceeded the Limits of Quantification (LOQ = 4 µg kg^−1^) in the treated group. The litter matrix had statistically significant differences between the study groups. The persistence of tylosin residues in the litter of birds at day 42 was 290.16 µg kg^−1^ in the treated group (A) and 9.35 µg kg^−1^ in the adjacent untreated group (B.1). **Conclusions:** The results indicate that exposure distance influences tylosin residue dissemination.

## 1. Introduction

Antibiotics are considered contaminants of emerging concern (CEC) due to their cumulative toxic effects on various organisms. Also, the persistence of these contaminants has been linked to the development of antibiotic-resistant bacteria, especially the impact of low concentrations of antimicrobial residues on the microbial diversity of the ecosystem [1,2]. As a result of increased food production, agricultural activities have become one of the most important drivers in releasing emerging contaminants into the environment, with antibiotic residues from livestock waste being a major contaminant with cumulative capabilities, potentially polluting the environment [2,3]. In poultry, antimicrobials (AMs) have been widely used to treat infectious diseases. In some cases, they have been prescribed preventively and as growth promoters, often administered through feed or drinking water [4]. However, this practice risks reintroducing antimicrobial residues into the food chain, contaminating poultry products, byproducts, and the surrounding environment [5,6].

Tylosin, a macrolide antimicrobial widely used in poultry production, is a key therapeutic agent for treating infections caused by *Mycoplasma gallisepticum*, *Clostridium* spp. and *Streptococcus pyogenes*. Despite its veterinary importance, research on tylosin persistence in the environment and its contribution to antimicrobial resistance remains limited compared to other AM classes. This gap is concerning because tylosin residues can accumulate in poultry litter composed of droppings, feed waste, feathers, wastewater, and bedding materials, which are highly susceptible to AM contamination [7,8,9]. Several studies have been reported on the detection of residues of this drug in edible and non-edible products of the poultry industry, such as muscle, liver, feathers, litter, and droppings, among others [9,10,11]. This is the case of the studies by Pokrant et al. [12] and Paranhos et al. [9], where they validated a methodology to detect residues of this AM in poultry litter and droppings, respectively. They subsequently tested the method in different samples and found residues in these matrices. Critically, 17–90% of administered AMs are excreted in the original form or as active metabolites, depending on the compound and species [13]. Once excreted, AM residues can permeate the broader production environment, including housing facilities, slaughterhouses, and waste management systems. Such contamination is of critical concern, as these components are intricately linked to the food production chain, creating pathways for residues to re-enter human and ecological systems [14,15,16,17].

Beyond direct contamination pathways, airborne particulate matter and bioaerosols from poultry facilities have been documented to carry AM residues, bacteria, and resistance genes. McEachran et al. [18] detected tylosin residues in particulate matter at concentrations of approximately 340 µg kg^−1^ at distances of 10–20 m. Pollutant concentrations in bioaerosols, including AM residues, tend to decrease with increasing distance from the emission source [8]. Recent controlled studies also revealed that antimicrobial residues, such as oxytetracycline and sulfachloropyridazine, can disseminate into untreated pens under certain conditions [19,20]. Although the extent of this dissemination appears limited, the phenomenon requires further investigation to elucidate the full implications. In the case of tylosin, although the depletion of this macrolide in chicken droppings has been studied [19], its ability to disseminate in a controlled poultry environment has not been studied; the distance of exposure is a variable to be considered to understand the behavior of antimicrobial residues in productive environments.

Environmental risks are further amplified by the agricultural reuse of poultry litter and manure, commonly employed as fertilizers or feed supplements [21,22]. These matrices, excluded from regulatory oversight due to their non-human consumption status, have been shown to introduce AM residues into soil. Rashid et al. [23] identified macrolide residues in soil amended with poultry manure, with tylosin detected in 85% of manure samples and 50% of soil samples. Concentrations ranged from 1.65 to 72.8 µg kg^−1^ in manure to 1.65–9.9 µg kg^−1^ in soil, demonstrating the transfer of AMs into environmental matrices and their persistence at ecologically relevant levels. These findings underscore the potential for the agricultural reuse of poultry waste to perpetuate antimicrobial resistance by maintaining selective pressure on soil microbial communities. Therefore, the potential dissemination of antimicrobial residues within productive environments reinforces their accumulation in non-edible by-products such as poultry litter, introducing these emerging contaminants into the food chain, affecting the environment as well as human and animal health, but mainly contributing to the development of antimicrobial resistance by microorganisms.

Although the existing literature demonstrates the prolonged persistence of veterinary antimicrobial residues in the environment, the role of broiler droppings, litter [24,25,26], and exposure distance in dissemination within poultry systems remains poorly understood. Moreover, not all classes of antimicrobials used in poultry production, such as tylosin, have been thoroughly studied in this context. This knowledge gap limits the development of effective strategies to mitigate the risks associated with antimicrobial use. To address this, the present study aimed to evaluate the effect of exposure distance on the spread of tylosin residues from the litter and droppings of therapeutically treated broiler chickens to untreated birds, providing insights into the role of poultry matrices in dissemination and contributing to the control of associated risks.

## 2. Results

### 2.1. Validation and Optimization of Chemical Analytical Method

The HPLC-MS/MS method for the detection of tylosin in litter and droppings complied with all parameters established by the Commission Implementing Regulation (European Union) 2021/808/EC [27] and VICH GL2 [28] guidelines. No interferences were observed at the retention time of tylosin, corresponding to 3.55 min for the quantifier and qualifier ion and 3.65 min for the internal standard erythromycin-N-methyl-13C-D3 (Figure 1). The limit of detection (LOD) and quantification (LOQ) of the analytical method were 1 and 4 µg kg⁻^1^, respectively. Additionally, the method demonstrated precision, with a relative standard deviation (RSD) below 25% for reproducibility and 16.6% for repeatability, with a recovery rate close to 100%, as detailed in Table 1. The method also proved to be linear and robust, with an R^2^ ≥ 0.96 and >0.05 *p*-value using the Mandel test, obtaining a standard deviation expressed as the mass/mass lower than that of precision (0.04 µg kg⁻^1^ versus 2.03 µg kg⁻^1^ for litter; 0.0001 µg kg⁻^1^ versus 0.0010 µg kg⁻^1^ for droppings). Finally, the matrix effect showed an RSD of 11.59% in litter and 10.48% for droppings.

The analyte in litter ranged between 41 and 46 µg kg⁻^1^ during the first and second weeks. However, by the third-week post-fortification, the concentration decreased to 35.14 µg kg⁻^1^. In droppings, the concentrations were consistent between 38 and 40 µg kg⁻^1^ during the first and second weeks but declined to 36.40 µg kg⁻^1^ by the third week.

### 2.2. Detection and Quantification of Tylosin Residues in Litter and Droppings in Poultry

The tylosin concentration from the bird litter of the treated group (A) started with an average of 1313.88 μg kg^−1^ on day 3 post-treatment. Concentrations above the limit of quantification (LOQ = 4 µg kg^−1^) were identified throughout the whole production cycle 42 days). In the untreated group directly next to the treated group (B.1), the concentrations started at 25.89 µg kg^−1^ on day 3 post-treatment, decreasing gradually but still exceeding the LOQ through day 18 post-treatment. In the sentinel group (B.2), separated by a 30 cm distance from the treated group, concentrations were 11.41 µg kg^−1^ on day 3 post-treatment. However, at the end of the productive cycle, the concentrations were lower than the limit of detection (LOD = 1 µg kg^−1^), which meant that no concentrations were detected in these matrices (Table 2).

Tylosin concentrations in droppings from group A averaged at 8.67 μg kg^−1^ on the third day after treatment, with trace concentrations found between day 6 and 9 post-treatment. Concentrations decreased below the LOD during the production cycle. For both sentinel groups, B.1 and B.2, concentrations were not quantified during the production cycle (Table 2).

For the control group (C), no concentrations were detected for both litter and droppings.

### 2.3. Determination of the Dissemination of Tylosin Residues in the Poultry Environment

The non-parametric Kruskal–Wallis test was used to evaluate the differences between the medians of the groups under study, determining statistically significant differences between the tylosin concentrations detected in the litter of the treated group (A), the adjacent group (B.1), and the group distanced 30 cm from the treated group (B.2) (*p*-value = 2.2 × 10^−16^). The eta-squared estimate was 0.8 with a 95% confidence level, demonstrating a moderate effect of distance on detected concentrations of tylosin residues, identifying that the shorter the distance of exposure to animals treated with tylosin, the higher the concentrations of residues disseminated in the production environment. Subsequently, Dunn’s test was used to evaluate which groups showed these differences, making the following three comparisons: group A–group B.1 (*p*-value = 1.241669 × 10^−8^), group A–group B.2 (*p*-value = 7.517277 × 10^−18^), and group B.1–group B.2 (*p*-value = 7.010577 × 10^−4^), indicating that all the medians are different from each other (Figure 2).

## 3. Discussion

This study detected tylosin residue concentrations in broiler litter and droppings using HPLC-MS/MS, which complies with the requirements of the Commission Decision 2021/808/EC [27], the Food and Drug Administration (FDA), and VICH GL2 [28] guidelines, confirming the accuracy and reliability for the detection and quantification of tylosin residues in broiler chicken droppings and litter.

After antimicrobial treatment, we detected tylosin residue concentrations in broiler litter and droppings and the dissemination of these residues to the untreated sentinel groups. Group A presented higher concentrations for both matrices, followed by group B.1 and then group B.2. These findings could be explained by the proximity of B.1 to the treated group, which likely resulted in increased exposure to contaminated dust and litter particles, emphasizing the critical role of physical separation or barriers in mitigating the risk of cross-contamination.

In droppings, residues were quantified only in the first sampling in group A (8.67 µg kg^−1^). These concentrations decreased during the study, falling below the detection limit by the end of the experiment in every group. These low concentrations could be explained by tylosin’s low oral bioavailability (13.7%) and short half-life (1.5–2 h) [29], suggesting that the most excretion occurred during the treatment and before the first sampling point. Previous studies reported variable tylosin concentrations in manure (1.65 to 13740 µg kg^−1^), though in field conditions with uncontrolled environments and different methodologies that may explain the variability in results [23,30]. Pokrant et al. [12] performed a study using a similar method and reported 104.66 µg kg^−1^ in droppings 5 days post-treatment. This difference could be explained by variations in the treatment duration (7 versus 5 days) and sampling method, where combined samples from different groups could have led to cross-contamination.

Litter showed higher concentrations due to the accumulative effect of different elements and the organic composition of the litter, which facilitates macrolide accumulation [8,31]. These findings align with Pokrant et al. [19], who reported ten times higher oxytetracycline concentrations in litter compared to droppings. Additionally, antimicrobial residues may also be found in other litter components, such as feathers, where tylosin concentrations of 9246.53 µg kg^−1^ have been reported 4 days post-treatment, decreasing to 356.59 µg kg^−1^ by day 15 [32].

Limited controlled studies exist on tylosin residues in poultry litter; however, reported concentrations have ranged from 135 to 597 µg kg^−1^ [9,11]. Our study detected concentrations from 1313.88 to 290.16 µg kg^−1^ in litter between days 3 and 18 post-treatment. These results exceeded those reports in field studies on the first sampling points, likely due to differences in sampling conditions and animal management.

Tylosin’s residue dissemination to sentinel groups was detected, suggesting that bird movement, such as dust bathing, facilitates the dispersion of litter components [33]. Considering the above, chicken litter plays a fundamental role in the persistence of these residues. This matrix is composed of droppings, feed, shavings, and feathers, and it has been reported that the prevalence of antibiotic-resistant bacteria can exceed 60% for specific microorganisms. These include *Escherichia coli* and *Enterococci*, which show multi-resistance to different families of antimicrobials [34]. Kruskal–Wallis and Dunn’s tests showed significant differences between the groups, with higher tylosin concentrations in the adjacent group (B.1) compared to the 30 cm distant group (B.2). This suggests that distance affects dissemination, with residues decreasing as separation increases. Although our study distances do not reflect commercial settings, they provide insight into future research considering larger-scale production systems. It is relevant to continue investigating both the persistence and dissemination of residues in larger-scale production conditions that are more representative of the industry, considering other factors such as greater distances of exposure, a greater number of animals, and the administration of antimicrobials in feed, among other factors.

Despite group differences, all concentrations decreased over time, which was confirmed through linear regression analysis. Group A (treatment) exhibited the steepest decline, likely due to the cessation of excretion. At the same time, sentinel group B.1 may have experienced slower reductions due to their continuous exposure to group A. Pokrant et al. [19] observed a 43% oxytetracycline reduction in litter, compared to 78% for tylosin in this study, likely due to this lower environmental persistence of antimicrobials [33,35]. Future studies should include more frequent sampling and individual fecal collections to minimize dilution effects and evaluate the excretion of tylosin. Also, post-euthanasia litter sampling could help determine the long-term persistence of tylosin in the matrix.

The persistence of antimicrobial residues can be a risk for both humans and animals that are exposed to them. Mainly because of the development of antimicrobial resistance, as pathogens are continuously exposed to these drugs, where selection pressure favors the expression and transmission of resistance genes, this causes difficulties in the treatment of infectious diseases in both humans and animals. On the other hand, exposure to residues of these drugs can also cause hypersensitivity reactions, carcinogenic effects, and imbalance in the intestinal microbiota, among other results [8,15].

Tylosin residues in poultry waste pose environmental concerns, as residues have been detected in fertilized soil [23], potentially contributing to antimicrobial resistance [36]. Tylosin may also induce cross-resistance to other macrolides, such as erythromycin [37].

This study shows that these byproducts are a potential source for disseminating antimicrobials in the production system. Further research under commercial conditions is necessary to assess tylosin’s persistence and dissemination, incorporating factors such as greater exposure distances, larger flock sizes, and feed-based antimicrobial administration.

## 4. Materials and Methods

The experimental design was carried out at the Animal Management Unit (UMA) located at the Faculty of Veterinary and Livestock Sciences of the University of Chile (FAVET), Santiago, Chile. Male Ross 308 broiler chickens were used as the study model. These were obtained from “Agrícola Chorombo” on their first day of life and were immediately transported to the facility where the study was conducted. The birds were received on the first day of life under controlled environmental conditions (ambient temperature: 27 °C; litter temperature: 30 °C). Conditions such as temperature (25 ± 5 °C) and humidity (50–60%) were regulated using heating and ventilation systems to maintain appropriate ranges according to the birds’ age requirements. The floor was covered with about 30 cm of sanitized wood shavings with a 5 to 20 mm particle size. The animals had ad libitum access to water and feed. The study was conducted in an Animal Handling Unit at the Faculty of Veterinary and Animal Sciences of the University of Chile.

Biosecurity measures for working with animals were implemented as suggested in the “Biosecurity and Associated Risks Standards Manual—Fondecyt (National Fund for Scientific and Technological Development)—CONICYT (National Commission for Scientific and Technological Research)” [38]. The handling and euthanasia of the animals were conducted in compliance with the guidelines of Law No. 20.380 on Animal Protection [39], Directive 2010/63/EU [40], and Regulation (EC) No. 1099/2009 [41]. The study was conducted under Certificate No. 22551–VET–UCH issued by the Institutional Committee for the Care and Use of Animals (CICUA) of the University of Chile.

### 4.1. Description of Experimental Animals

The study was conducted in two separate rooms: one designated as the control and the other as experimental. The first room contained a single pen for the control group (C), while the second room had three pens for experimental groups A, B.1, and B.2. Each pen measured 1 m^2^, and 40 animals were used, with 10 birds assigned to each group. The number of birds was determined based on the fact that broiler chickens can weigh up to 3.2 kg, and the “Poultry Industry Manual” of the United States Department of Agriculture (USDA) recommends a maximum live weight density of 34.372 kg/m^2^ [42]. The experimental design is shown in Figure 3.

The antibiotic tylosin phosphate 10% was used for the treatment, administered at a dose of 35 mg kg^−^^1^ of tylosin base every 24 h for five days, as indicated on the product label. Treatment began on day 20 of life and was administered orally using a Levin No. 6 orogastric tube (Cranberry International Sdn. Bhd., Shah Alam, Malaysia) to ensure the full dose was delivered to each animal. The vehicle used for the administration of antimicrobials was distilled water.

#### Sampling of Litter and Droppings

Samples of litter and droppings were collected from each group. A total of six samples were conducted on days 3, 6, 9, 12, 15, and 18 post-treatment. The pen was divided into nine quadrants using the grid method for the litter samples, as described in the IAEA report “Soil Sampling for Environmental Contaminants” (Figure 4) [43]. Approximately 20 g of litter was collected from each quadrant, resulting in a final 180 to 200 g sample per experimental group. The sampling order was as follows: group C, group B.2, group B.1, and group A to prevent the cross-contamination of untreated groups.

Dropping samples were collected in 50 mL polypropylene tubes. Cloacal stimulation was performed on each individual using sterile swabs to collect the samples. The samples from each group were pooled into a single tube, as the focus of this study was to evaluate the group-level concentrations rather than individual variations. All samples were obtained in triplicate and stored at −20 °C until processing.

### 4.2. Chemical Analysis

#### 4.2.1. Tylosin Standards

Tylosin tartrate (98% purity; CAS:1401-69-0) and erythromycin-N-methyl-13C-D3 (95% purity; CAS: 2378755-50-9) were used in this study, with the latter employed as the internal standard of the sample analysis. These standards were purchased from Toronto Research Chemicals (Toronto, ON, Canada). Stock solutions were prepared in methanol at 1000 µg mL⁻^1^, and intermediate or working solutions were diluted to 1000 ng mL⁻^1^ in the same solvent. These solutions were stored at −80 °C.

#### 4.2.2. Chemicals and Solvents

HPLC-grade water, acetonitrile, and methanol were purchased from LiChrosolv^®^ (MERCK KGaA, Darmstadt, Germany). Citric acid monohydrate (EMSURE^®^ ACS, Reag. Ph Eur, MERCK KGaA, Darmstadt, Germany), dibasic sodium phosphate (Na_2_HPO_4_, EMSURE^®^ ACS, Reag. Ph Eur, MERCK KGaA, Darmstadt, Germany), and disodium salt dihydrate of ethylenediaminetetraacetic acid (Na_2_EDTA, Titriplex^®^ III ACS, ISO, Reag. Ph Eur, MERCK KGaA, Darmstadt, Germany).

The 0.1 M EDTA/McIlvaine buffer (pH 4.0 ± 0.1) was used for tylosin extraction from matrices. Extracts were prepared by mixing 0.1 M citric acid, 0.2 M disodium phosphate, and Na_2_EDTA, followed by sonication and pH adjustment with citric acid or phosphate solutions.

Two solutions were prepared for chromatographic analysis. Eluent A was prepared by dissolving 2.0 mM of ammonium formate and 0.16% formic acid in water. Eluent B was prepared with the same concentration of ammonium formate and formic acid in methanol.

#### 4.2.3. Sample Preparation

The extraction method implemented by Yévenes et al. [44] and Pokrant et al. [12] was followed. Bedding and droppings were homogenized using a blender and a sterile wooden stick. A 1 ± 0.01 g sample was weighed, spiked with intermediate solution standards, and subsequently subjected to extraction. The samples were enriched with an 8 mL McIlvaine-EDTA buffer (pH 4.0 ± 0.1) and 2 mL acetonitrile, followed by 10 min of agitation, 5 min of sonication, and 10 min of centrifugation at 4000–5000 rpm. Filtration was performed using glass microfiber Whatman™ filter paper of grade GF/A (1.6 µm) (MERCK) and SPE Oasis HLB columns (Supelco, MERCK KGaA, Darmstadt, Germany). Solvents were evaporated under nitrogen flow in a water bath at 40–50 °C, and the sample was reconstituted with 200 µL of methanol and 300 µL of water. Finally, the samples were agitated for 5 min, sonicated for 5 min, centrifuged for 5 min at 1700 rpm, and transferred to an amber glass vial using a syringe with Millipore filters (Millex^®^, Merck KGaA, Burlington, MA, USA).

This chemical analysis was performed at the Laboratory of Veterinary Pharmacology (FARMAVET) of the Faculty of Veterinary and Animal Sciences at the University of Chile, which is accredited under normative standards.

#### 4.2.4. LC-MS/MS Analysis

For tylosin detection, the ACQUITY UPLC^®^ I-Class System (Waters™, Milford, MA, USA), coupled with a Xevo TQ-S micro triple quadrupole mass spectrometer (Waters™, Milford, MA, USA) were used. Chromatographic separation was performed using an ACQUITY UPLC^®^ HSS T3 analytical column (1.8 µm, 2.1 × 100 mm; Waters™ Corp, USA), and the resulting data were processed with MassLynx™ software version 4.2 and TargetLynx^®^ xs software version 4.2.

The chromatographic conditions consisted of a sample injection volume of 10 µL, a mobile phase gradient flow rate of 300 µL/min, and a chromatographic column temperature of 40 °C. The percentage of each mobile phase injected over time is detailed in Table 3. The mass spectrometer operated in the positive ion mode with the following instrument conditions: a capillary voltage of 3.00 kV, source temperature at 150 °C, desolvation gas flow of 1000 L/h with a desolvation temperature of 600 °C, and a sample cone gas flow of 10 L/h. Specific mass spectrometric settings for the analytes are detailed in Table 4.

Tylosin was confirmed in the experimental samples by meeting the following criteria: retention time and relative retention time with a variation not exceeding ±2.5%, a signal-to-noise ratio of at least 3:1, and a relative deviation of the ion ratio within ±40%. Once confirmed, quantification was performed using calibration curves derived from fortified blank samples spiked at different, equidistant levels and processed alongside the experimental samples.

#### 4.2.5. Analytical Method Validation/Verification

The analytical method was validated for poultry litter and droppings according to the recommendations from Commission Decision 2021/808/EC [27] of the Food and Drug Administration (FDA) and VICH GL2 [28]. For this purpose, the following parameters were evaluated:Limit of detection (LOD): Twenty fortified samples were analyzed at the selected LODs, with a signal-to-noise ratio of ≥3:1. A coefficient of variation (CV%) of <25% was accepted.Limit of quantification (LOQ): Twenty fortified samples at the LOD were analyzed, and the standard deviation was determined, which was then multiplied by 1.64 and added to the LOD. The signal-to-noise ratio for this parameter must be ≥10:1 to be accepted.Retention time (TR): Six samples with a tylosin standard were analyzed to evaluate the TR of each, with an accepted coefficient of variation of <1%.Specificity: Twenty blank samples were evaluated to determine if there were any interferents in the TR.Linearity: Three calibration curves were created using five different concentrations, with the lowest concentration at LOD. A determination coefficient (R^2^) ≥ 0.95 and a *p*-value >0.05 using the Mandel test was required for acceptance.Precision and recovery: To achieve precision, two parameters were evaluated: intra-laboratory reproducibility and repeatability. Six calibration curves at 25, 50, and 75 µg kg^−1^ were analyzed for each parameter on three days. Repeatability was assessed by performing all analyses under identical conditions by the same analyst, while intra-laboratory reproducibility involved analyses conducted by different analysts under varying conditions. Recovery was evaluated based on these samples, and this was achieved when the mean recovery percentage was between −20 and 20% on each level.Matrix effect: A chemical extraction was performed on 20 blank samples fortified at the end of the extraction process. The signal obtained from these samples was compared to pure standard injections.Ruggedness: This was evaluated by modifying three conditions: the amount of EDTA/McIlvaine buffer added to the sample initially (8 mL to 4 mL of buffer), the centrifugation time at the beginning of the extraction process (10 to 5 min), and the volume used to wash the extraction column (5 mL to 2 mL of HPLC water). The method was considered robust when the standard deviation of the modified samples was equal to or lower than intra-laboratory reproducibility.Stability: fifteen chicken litter and dropping samples were fortified at 50 µg kg^−1^ and stored at −20 °C. Five samples were analyzed after 1 week, another five samples were analyzed after 2 weeks, and the remaining five were analyzed after 3 weeks.

#### 4.2.6. Statistical Analysis

To assess the effect of distance on the residue concentrations observed across different groups, the normality of the data was first evaluated using the Shapiro–Wilk test. Homoscedasticity was assessed using Levene’s test. As the data did not meet these assumptions, the non-parametric Kruskal–Wallis test was applied, considering a *p*-value < 0.05 and a 95% confidence interval to determine if there were statistically significant differences between the medians of tylosin concentrations within the study groups. This aimed to evaluate whether the exposure distance affected the antimicrobial residue concentration in broiler litter and droppings. Subsequently, Dunn’s test was performed to assess the differences between the medians of concentrations for each group. The matrices were evaluated individually. Statistical analyses were carried out using R version 4.3.2 and RStudio version 2023.09.1+494.

## 5. Conclusions

Tylosin is an antimicrobial that can spread within the production environment through the matrices studied. Litter presented the highest concentrations in the samples taken, and the presence of residues decreased as the exposure distance from the pens increased. This implies that these matrices are a potential route for disseminating AM residues, posing a hazard to public health. Broiler chicken droppings and litter are byproducts of the poultry production process and serve as indicators of AM residues in the poultry farm. These matrices are easily accessible, and their collection is non-invasive for the animals, allowing for the continuous assessment of the presence of AM residues in the production environment.

## Figures and Tables

**Figure 1 antibiotics-14-00477-f001:**
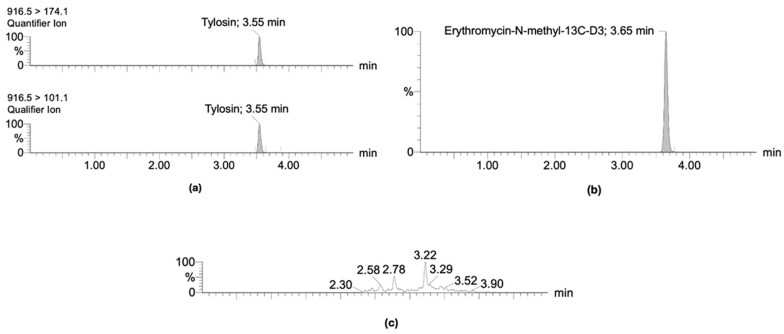
(**a**) Chromatograms of tylosin quantifier and qualifier ion. (**b**) Chromatogram of internal standard erythromycin-N-methyl-13C-D3. (**c**) Chromatogram of litter sample free of tylosin residues.

**Figure 2 antibiotics-14-00477-f002:**
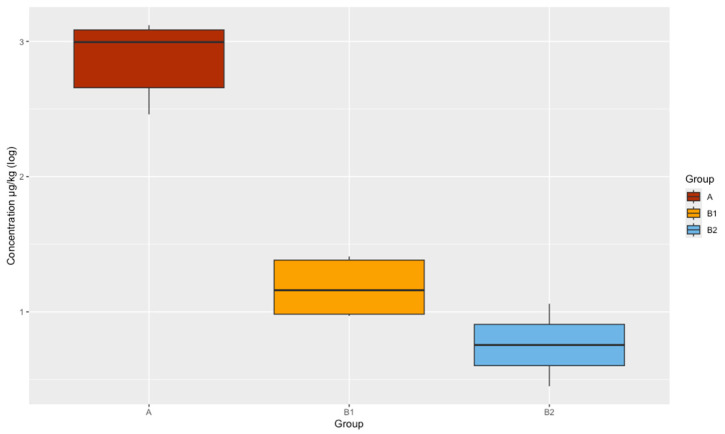
The boxplot of tylosin concentrations by experimental groups transformed into a natural logarithm in the broiler chicken litter matrix. There is evidence of differences between medians (Kruskal–Wallis; *p*-value < 0.05).

**Figure 3 antibiotics-14-00477-f003:**
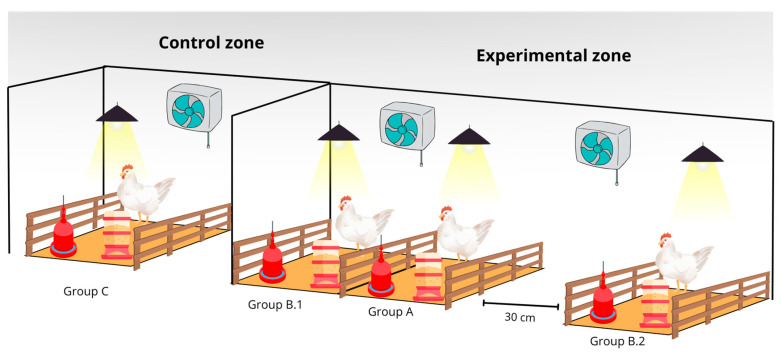
The experimental design of the study. Each group consisted of 10 broiler chickens. The control group was housed separately, while group A received a therapeutic dose of tylosin. Groups B.1 and B.2 served as sentinel groups, with group B.2 placed at 30 cm.

**Figure 4 antibiotics-14-00477-f004:**
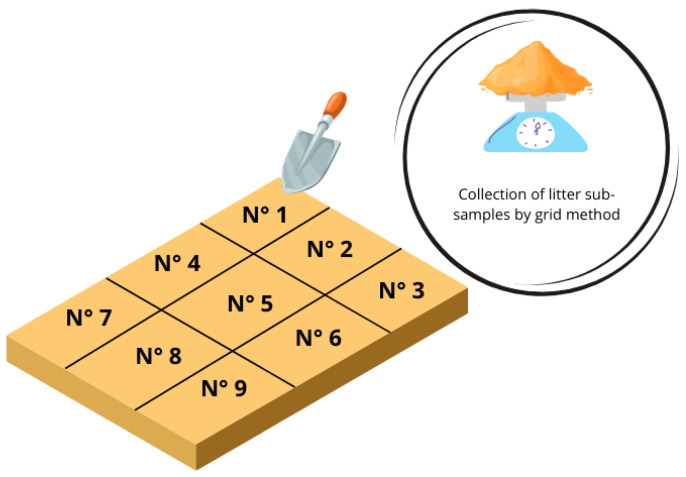
The sampling of poultry litter using the grid method. Equal samples were obtained by the sampling point and subsequently homogenized.

**Table 1 antibiotics-14-00477-t001:** Precision and recovery results of the validation method for the detection of tylosin in chicken droppings and litter.

Matrix	Fortified Concentration (µg kg^−1^)	Repeatability (RSD %) *	Reproducibility (RSD %)	Recovery (RSD%)
Droppings	25	3.01	4.67	98.34
	50	2.88	4.52	101.66
	75	0.99	1.54	99.45
Litter	25	4.17	4.61	97.11
	50	3.96	4.35	102.89
	75	1.37	1.51	99.04

* Relative standard deviation expressed as percentage.

**Table 2 antibiotics-14-00477-t002:** Tylosin concentrations detected in the litter and droppings of treated and untreated birds from each experimental group by the number of post-treatment days.

Tylosin Concentrations (µg·kg^−1^ ww)							
Group	Matrix	Day 3	Day 6	Day 9	Day 12	Day 15	Day 18
A	Litter	1313.88	1284.94	1015.09	955.42	351.61	290.16
	Droppings	8.67	<LOQ **	<LOQ	<LOD *	<LOD	<LOD
B.1	Litter	25.89	24.95	21.38	9.71	9.51	9.35
	Droppings	<LOQ	<LOD	<LOD	<LOD	<LOD	<LOD
B.2	Litter	11.41	<LOQ	<LOQ	<LOQ	<LOD	<LOD
	Droppings	<LOD	<LOD	<LOD	<LOD	<LOD	<LOD

ww: wet weight; * <LOD: below the limit of detection for the technique (droppings and litter 1 µg kg^−1^); ** LOQ: below the limit of quantification for the technique (droppings and litter 4 µg kg^−1^).

**Table 3 antibiotics-14-00477-t003:** Mobile phase gradient flow.

Time (Minutes)	Solution A (%)	Solution B (%)
0.00	98	2
1.00	98	2
2.00	55	45
3.10	25	75
3.30	98	2
5.50	98	2
7.00	98	2

**Table 4 antibiotics-14-00477-t004:** Specific mass spectrometric conditions.

Analyte	Precursor Ion (*m*/*z*)	Productor Ion (*m*/*z*)	Dwell Time (s)	Cone Voltage	Collision Energy
Tylosin	916.50	174.10 ^1^	0.009	45.00	40.00
		101.10 ^2^	0.009	45.00	45.00
Erythromycin-N-methyl-13C-D3	738.40	162.10	0.009	20.00	30.00

^1^ Quantifier ion; ^2^ qualifier ion.

## Data Availability

The data presented in this study are available in the article.

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
