# Peer review of "Dissemination of Tylosin Residues in the Poultry Environment: Evaluating Litter and Droppings as Sources of Risk"

_antibiotics, 2025, doi:10.3390/antibiotics14050477_

Round 1
Reviewer 1 Report
Comments and Suggestions for Authors
Overall this a very well written manuscript. A few minor comments and edits.
Introduction
Line 48 – should “described” be “prescribed”?
First three paragraphs should be combined into one paragraph.
Lines 52-65 should be one paragraph
Results
Line 132 – you have a question mark for the number of days
I appreciate the use of LOD and LOQ in the tables and text but I'm not sure it was very clear in the text that there were samples in which tylosin was detected but not quantified. I think this needs to be more clearly stated in the text
Methods
Line 231 – what do “Fondecyt” and “CONICYT” stand for?
Reviewer 2 Report
Comments and Suggestions for Authors
The study addresses a highly relevant topic in veterinary pharmacology and environmental health, focusing on disseminating tylosin residues within poultry environments. The manuscript is well-structured, employs rigorous analytical methods (HPLC-MS/MS), and contributes novel data on antimicrobial dissemination within broiler systems. The public health relevance—especially in the context of antimicrobial resistance (AMR)—aligns well with the scope of MDPI Antibiotics. Language and presentation are generally clear and professional. However, there are areas for improvement in both presentation and scientific clarity, particularly regarding the interpretation of results, depth of discussion, and literature contextualization.
Please check and correct the use of units "μg kg-1" and "μg/kg" throughout the article.
Page 4, Line 132: "(? days)" is not understood. Please check.
The introduction should highlight what gaps this study uniquely addresses compared to similar work (e.g., Pokrant et al. [30], [32]). Emphasize how this research extends previous findings by quantifying tylosin in different exposure distances and discuss its real-world implications more clearly.
The distance between pens (30 cm) is narrow and may not reflect commercial poultry settings. So, please discuss this limitation more critically and suggest how this could be scaled for field applications.
The non-parametric tests (Kruskal-Wallis, Dunn’s) are appropriately used due to data distribution. However, the effect sizes or confidence intervals for these differences are not reported. So, please include effect size estimates (e.g., eta squared for Kruskal-Wallis) to quantify the magnitude of differences better.
The Discussion mostly reiterates results without deeply exploring mechanisms (e.g., dust particle transport, environmental factors). The public health implications of these findings (e.g., impact on AMR reservoirs in soil or food chains) could be expanded. So, please broaden the discussion to include One Health perspectives, explicitly connecting environmental dissemination to potential human and animal health risks.
Figure 2: Add units (µg/kg) to axes for clarity.
Reviewer 3 Report
Comments and Suggestions for Authors
Line 73: It is recommended to include specific background information related to tylosin in the introduction. Since data on other antibiotics are mentioned, are there previous studies reporting its behavior in similar matrices to better contextualize the choice of this molecule?
Line 129: The presentation of the results should be improved, as currently the data are listed without interpretation. Consider comparing between groups and highlighting the biological or environmental implications of the findings.
Line 161 (Discussion): Overall, the writing style tends to be overly narrative. It is suggested to revise the section to increase the interpretative analysis of the results. For example, instead of stating: “Group B.1 had higher tylosin concentrations than B.2”, you could rephrase: “The proximity of B.1 to the treated group likely increased its exposure to contaminated dust and litter particles, highlighting the importance of physical separation or barriers to minimize cross-contamination.”
Line 220: Please include specific information about the geographical location of the study (city, country, institution), as this is relevant for reproducibility and environmental context.
Line 222: The phrase "Chorombo (or a similar source)" should be replaced with a precise description of the actual supplier or, alternatively, the criteria used to define a “similar source,” to avoid methodological ambiguity.
Line 224: Please verify whether the reported temperature (25 ±â€¯5 °C) is appropriate for day-old chicks, as it may be considered low for that developmental stage.
Line 225: It is suggested to clarify that environmental conditions such as temperature and humidity were progressively adjusted according to the animals’ age, as part of standard broiler management practices.
Line 227: Include specific details regarding litter management, such as frequency of replacement or strategies for controlling moisture and ammonia levels, as these may influence residue accumulation.
Figure 3: It is recommended to add a complementary diagram illustrating the grid-based subdivision of the pen floor used for sampling, as described in the methodology.
Line 248: Since tylosin is the central focus of the study, it should be described in a separate subsection of the materials section, including characteristics of the active compound, batch number, and concentration of the inoculum.
Line 251: Please specify the vehicle used for tylosin solution preparation (e.g., distilled water, saline, or a commercial diluent), as it may affect absorption and compound stability.
Line 253: If fecal samples were pooled into a single tube per group, individual traceability is lost. This should be clarified to ensure transparency regarding sampling limitations.
Line 263: Confirm whether only one pooled sample per group and time point was collected without technical replicates. If so, consider discussing the limitation in terms of statistical power.
Line 286: To ensure reproducibility of the extraction protocol, it is necessary to detail critical parameters such as the duration of each step, centrifugation speed (g or rpm), sonication time and frequency, and the volume of solvent used at each stage.
Line 291: There is an inconsistency in units“pH 4.0 ± 0.1 M” is incorrect, as pH is not expressed in molarity. Please verify and correct this apparent typographical error.
Line 239: Please confirm whether Regulation 2021/808/EC is from the European Commission, as it is not a guideline issued by the FDA. This clarification is important for correct regulatory attribution.
